

# Variations in Turbulence Characteristics of Head Fires Induced by Background Wind and Fuel Loading Levels

Chuanying Lin [1], Xingdong Li [1,2], Hongyang Zhao [1], and Dandan Li [1]

[1]College of Mechanical and Electrical Engineering, Northeast Forestry University, No. 26, Hexing Road, Harbin 150040 Heilongjiang,China
[2]Northern Forest Fire Management Key Laboratory of State Forestry and Grassland Administration, Northeast Forestry University, No. 26, Hexing Road, Harbin 150040 Heilongjiang,China

**Correspondence:** Xingdong Li  (lixd@nefu.edu.cn)

**Abstract.** Turbulence caused by wildfires can significantly alter the meteorological conditions in the fire area, which is a primary cause of the instability and unpredictability of fire behavior. Observing the differences in turbulence characteristics and propagation patterns caused by variations in fuel and background atmospheric conditions is crucial for the early warning of extreme fire behavior resulting from high-intensity fire-atmosphere interactions based on pre-fire information. This study, based

on high-density measurements of fire-induced turbulence, examines how the differences in turbulence characteristics between the pre-burn, burning, and post-burn periods are influenced by variations in background mean wind speed and fuel load. It also investigates the impact on the spatial heterogeneity of turbulence characteristics, particularly their changes with increasing distance from the ignition boundary. Turbulence characteristics are quantified by parameters including instantaneous wind speed fluctuations, turbulent kinetic energy, heat flux, and momentum flux. The analysis indicates that the spatial heterogeneity

of turbulence characteristics is weakened with increasing background wind speed; an increase in fuel load will result in a prolonged disturbance of atmospheric conditions by the fire; both increases in background wind speed and fuel load lead to the changes in the dominant components of certain parameters.

## 1 Introduction

The high instability and unpredictability of wildfire behavior often lead to the ineffectiveness of firefighting measures imple-

15 mented by wildfire managers, resulting in significant loss of life and property (Tedim, F. et al., 2018). The wildfire triangle consists of vegetation, terrain, and meteorological conditions (Oliveras, I. et al., 2009). Relatively speaking, the unpredictability of wildfire behavior largely depends on meteorological conditions: vegetation variables such as load, type, and moisture content typically exhibit low spatial and temporal variability; while terrain conditions, such as slope and elevation, can be effectively obtained and used in predictive models. However, complex interactions and feedback between the fire and atmospheric

dynamics, which produce vortices and turbulence that are often difficult to capture(Potter, B.E., 2012; Heilman, W.E., 2023; Potter, B.E., 2012), can easily lead to abrupt changes in fire behavior. Therefore, assessing the characteristics of fire-induced turbulence and exploring the factors influencing fire-atmosphere interactions are crucial for understanding the instability of fire behavior and improving its predictability(Sun, R. et al., 2009).



In recent years, field measurement data from burn experiments of various scales have provided a crucial foundation for studying fire-atmosphere interactions. In 2006 and 2013, Clements et al. conducted the FireFlux 1 and FireFlux 2 experiments on a $40 ha$ native tallgrass prairie at the Houston Coastal Center in Texas(Clements, C. B. et al., 2007, 2019). Three-dimensional (3D) sonic anemometers were used, installed on towers at heights of 10 $m$ and 43 $m$, to measure turbulence during the fires. In March 2018, Finney et al. conducted ignition experiments on 12 mowed wheat fields in Fielder, New Zealand, each ranging from 1 to 3 $ha$(Finney, M. A. et al., 2018). A 10-$m$-high tower was placed in each plot, with sonic anemometers installed at 2 $m$, 5 $m$, and 10 $m$, to measure changes in atmospheric turbulence as the fire line passed. Between 2008 and 2010, Seto carried out 4 field experiments under different fuel and terrain conditions, with sizes ranging from 1 to 25 $ha$. Each experiment involved a sonic anemometer installed at a fixed height within the burn area to capture turbulence states(Seto, D. et al., 2013). Ottmar reported 16 prescribed burn experiments in the southeastern United States, covering areas from 2 to 828 $ha$ and including two types of vegetation cover: forest and non-forest(Ottmar, R. D. et al., 2015). Additionally,The United States Department of Defense (DoD) Strategic Environmental Research and Development Program (SERDP)funded project:Multi-scale Analyses of Wildland Fire Combustion Processes in Open-canopied Forests using Coupled and Iteratively Informed Laboratory-,Field-,and Model-based Approaches (RC-2641) conducted a large-scale (management-scale) field experiment during an operational prescribed burn on 13 March 2019 within the Brendan T. Byrne State Forest, New Lisbon, New Jersey, USA(Gallagher, M. R. et al., 2023). While these large-scale ignition experiments closely approximate real fire conditions, the low density of meteorological towers limits the spatial resolution of turbulence information captured. Consequently, the microscale fire-atmosphere interaction processes and spatial distribution characteristics of turbulence are often overlooked.

Small-scale combustion experiments are typically conducted under laboratory conditions. A series of burn bed-scale ignition experiments were performed by Finney et al. at the Missoula Fire Sciences Laboratory (MFSL) to examine how turbulence structures affect fire spread on very small spatial and temporal scales (approximately $10^{-2}m$, $10^{-1}s$) (Finney, M. A. et al., 2015). A portable wind tunnel, which can be used in both laboratory and field settings, was developed and implemented by Di Cristina, aiming to connect laboratory wildfire experiment results with those conducted in natural conditions (Di Cristina, G. et al., 2015). Studies on turbulent fire vortices, conducted at the laboratory scale, were also reported by Forthofer and Goodrick (Forthofer, J. M. et al., 2011). The advantage of these laboratory studies lies in the higher density of monitoring sensors compared to field experiments. However, the characteristics of fire-induced turbulence observed in these controlled conditions are challenging to extrapolate to real-world wildfire environments.

In comparison to large-scale field experiments and laboratory studies, the 35 prescribed fire experiments funded by SERDP of DoD, conducted in the New Jersey Pine Barrens from 2018 to 2019, effectively addressed the limitations of both methods(Ottmar, R. D. et al., 2015). These experiments covered an area of 100 $m^2$, with 16 3D sonic anemometers distributed evenly throughout the burned area(Clark, K. L. et al., 2023). This data provides strong support for evaluating the spatiotemporal changes in turbulence above and near the fire front within the burning area.

Based on existing data, the study of fire-atmosphere interactions has been extensive. One major area of focus is the heterogeneity of these interactions before, during, and after the fire front passes, as well as the heterogeneity at different locations within the fire field. From two field experiments conducted in tall grass fuels, it was observed that the turbulence intensity





during the FFP within the fire was 4-5 times that of the ambient turbulence intensity. After the FFP, it was noted that the turbulent kinetic energy rapidly decreased to slightly below the pre-fire level(Clements, C. B. et al., 2008). Heilman et al.examined the sweep-ejection dynamics that occurred before, during, and after the passage of a surface fire front during a prescribed fire experiment(Heilman, W. E. et al., 2021). Zhong, S. et al., based on the FireFlux 2 experiment, showed that fire-induced TKE disturbances are much stronger during Red Flag Warning conditions compared to less fire-prone atmospheric conditions,however it did not hold for momentum fluxes or friction velocity (Zhong, S. et al., 2024). Furthermore, using data from Burn 20 of the 35 prescribed fire experiments in the New Jersey Pine Barrens, Seitz confirmed that fire-induced atmospheric turbulence exhibited strong dependence on the downwind distance from the initial line fire and the relative position specific to the fire front as the surface fire spread through the burn plot(Seitz, J. et al., 2024).

The second type of research examines the impact of forest structure on fire-atmosphere interactions. Heilman et al.examined the effects of forest canopies on turbulence energy budgets, the skewness in turbulent velocity distributions, and stability–anisotropy variations before, during, and after fire-front-passage periods (Heilman, W. E. et al., 2017),the effects of spreading surface fires and forest overstory vegetation on turbulent heat and momentum fluxes from near the surface to near the top of the overstory vegetation(Heilman, W. E. et al., 2019) and the sweep–ejection dynamics that occur in response to turbulence regimes induced by wildland fires in forested and nonforested environments(Heilman, W. E. et al., 2021). Desai et al. synthesized observations from 4 field experiments (NJ2011, NJ2012, NJ2019, FireFlux) to identify and compare the fundamental coherent structures and processes in each case using turbulence momentum flux and TKE budget terms(Desai, A. et al., 2023). Wavelet analysis techniques were also used to explore the characteristic time scales of related patterns in temperature and turbulence flux measured during a wind-driven surface fire under the forest canopy (Desai, A. et al., 2024).

The third area of research evaluates the impact of topographical conditions within the wildfire triangle on fire-atmosphere interactions. In a 2008 burn experiment in a narrow valley of the Diablo Range in California, Seto et al. documented the formation of turbulent fire vortices. Their study demonstrated how sea breezes, thermally driven upslope winds, and fire-induced circulations interact, resulting in strong vertical wind shear that promotes turbulence and fire vortex formation(Seto, D. et al., 2011). Further studies by Seto et al. in 2010 involved multiple slope grass fire experiments near Dublin, California( Clements, C. B.et al., 2015). Charland et al.'s 2011 experiments focused on leeward downslope grass fires in San Jose, California( Charland, A. et al., 2013), and another prescribed grass fire experiment in 2012 was conducted on a simple slope at Fort Hunter Liggett in central California( Arreola Amaya, M. et al., 2020). These studies demonstrated that topographically induced slope flows and valley winds can interact with fire-induced flows, enhancing horizontal and vertical wind shear and promoting turbulence generation.

These studies have greatly enhanced our understanding of fire-atmosphere interactions. However, our knowledge of how vegetation and background meteorology, the other two components of the wildfire triangle, affect the spatiotemporal characteristics of fire-induced turbulence is still limited. Although an attempt by Clark et al. was made to evaluate the potential links between fuel consumption, fire behavior, atmospheric turbulence, and energy exchange, the robustness of their conclusions was limited by the low sensor density in their field experiments( Arreola Amaya, M. et al., 2020).





In this context, this paper will analyze variations in turbulence characteristics of head fires Induced by background wind and fuel loading levels, based on high-spatial-temporal resolution turbulence data collected during outdoor small-scale ($10m×10m$) experimental burns. Specifically, our research will address the following questions: Will the increase in background wind speed and fuel load amplify or reduce the differences in turbulence characteristics during the pre-burn, burn, and post-burn period? In the burn period, will the spatial heterogeneity of turbulence characteristics be enhanced or weakened? Will variations in background wind speed and fuel load levels modify the primary driving factors of certain turbulence characteristics? Turbulence characteristics here include the instability of horizontal and vertical wind speeds, turbulence intensity, and the exchange of turbulent heat and momentum between the burning area and the overlying atmosphere. The findings of this study can help wildfire managers to infer the intensity of fire-atmosphere interactions at known locations using pre-fire information. Additionally, high-risk areas within the fire zone where strong fire-atmosphere coupling is likely to occur can be identified. This knowledge will facilitate the early deployment of firefighting measures, thereby enhancing the effectiveness of wildfire management.

## 2  Data and Methods

### 2.1  Overview of burn experiments

The experimental dataset analyzed in this study was derived from a research project funded by the SERDP of the U.S.DoD. This project involved a low-intensity ground fire experiment with extensive instrumentation in the Silas Little Experimental Forest in New Lisbon, New Jersey, covering an area of $100 \, m^2$. The instrumentation setup at the experimental site is detailed in Figure 1. For a complete description of the experimrnt, see(Ottmar, R. D. et al., 2015). From March 2018 to June 2019, 35 combustion experiments were designed and conducted, encompassing a range of fuel conditions—including varying fuel loads, particle types, and bulk densities—and different meteorological conditions such as humidity, flow rates, and temperatures. To analyze the spatiotemporal characteristics of fire-induced turbulence resulting from different levels of vegetation load and background wind speed, data from these experiments were selected and two groups were formed, each with control experiments. It is important to note that the conditions of these field-based combustion experiments cannot be perfectly replicated. For example, even if two experiments are ignited simultaneously, the atmospheric conditions may differ because of the different locations of the fire sites. Additionally, if two experiments are ignited simultaneously at nearby locations, their interactions can substantially influence the results. Therefore, it is believed that for experiments conducted on adjacent dates within the same season, errors caused by seasonal factors such as temperature, humidity, atmospheric pressure, or fuel moisture content are considered to be within acceptable limits. Table 1 provides the detailed configuration parameters of the 4 selected combustion experiments, which were all downwind fires using loblolly pine needles as fuel, each resulting in the complete burning of the entire plot.





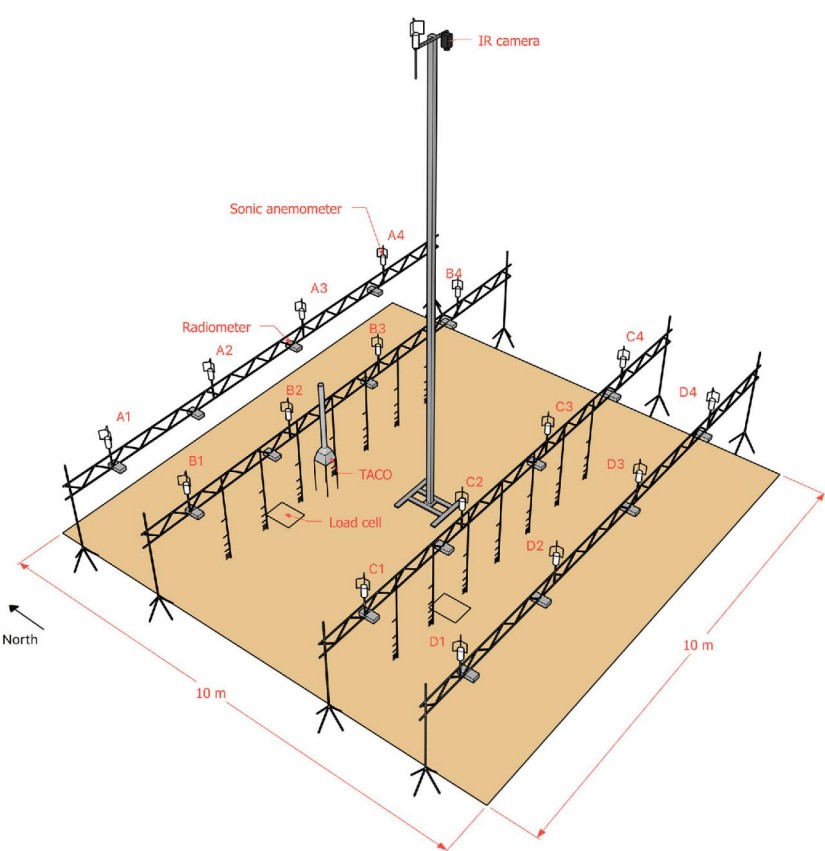

**Figure 1.** Instrumentation configuration diagram of the burn experiment site. 16 3D sonic anemometers (10Hz) are positioned at locations A1 to D4 in the diagram, at a height of 2.5 $m$ above the ground. Each anemometer is paired with a radiometer-spectral camera set. An infrared camera is mounted on a central pole in the field. Thermocouples are installed at various heights on the vertical columns of the B and C trusses. Adapted from(Seitz, J. et al., 2024)

## 2.2 Data analyses

To ensure data quality, we preprocessed the sonic data by removing samples outside the sonic anemometer's operating range of -50°C to 50°C and those with a diagnostic code (DIAG) not equal to 0. After this, we applied Seitz's criteria to define

the burn period for the quality-checked data(Seitz, J. et al., 2024). The burn period for each experiment began when any of the 16 sonic anemometers first recorded a temperature exceeding $\overline{T} + 8\delta$ ($\overline{T}$ and $\delta$ represent the mean and standard deviation of the temperature during the pre-burn period, respectively)and ended when the last of them recorded a temperature above the threshold. We defined the ten minutes before the infrared sensor first detected 300°C as the pre-burn period, and the ten minutes following the burn period as the post-burn period. After determining the three periods, we calculated the average wind

direction recorded by all the sonic anemometers during the pre-burn period to establish the prevailing wind direction for each





**Table 1.** Detailed configuration parameters of all selected burn experiments.

| Property | Units | Burn 2 | Burn 5 | Burn 11 | Burn 12 |
|---|---|---|---|---|---|
| Burn Data and start time | / | 3/6/2018 11:27:00 | 3/17/2018 12:42:50 | 5/11/2018 9:42:03 | 5/11/2018 11:55:00 |
| Mean ambient temperature | °C | 6.5 | 5.9 | 20.8 | 22 |
| Mean relative humidity | % | 42.8 | 25.9 | 56.1 | 50.5 |
| Mean wind speed | $m/s$ | 1.7 | 4.7 | 2 | 2.7 |
| Max wind speed | $m/s$ | 4.7 | 11.1 | 5.3 | 6.9 |
| Mean wind direction | Deg | 304 | 278 | 342 | 338 |
| Fuel loading | $Kg/m^2$ | 0.48 | 0.48 | 0.96 | 1.44 |
| Ignition side | / | east | west | west | west |
| Fuel dispersion | / | Naturally dispersed | Naturally dispersed | Artificially dispersed | Artificially dispersed |

The mean ambient temperature, mean relative humidity, mean wind speed, maximum wind speed, and mean wind direction were all measured from the control tower, which is 18 $m$ high and located 50 to 110 $m$ northwest of the burn site.

burn experiment. Subsequently, we projected the original horizontal wind vectors onto the prevailing wind direction and the direction perpendicular to it (pointing to the left) to obtain the streamwise component $u$ and the cross-stream component $v$.

The sonic anemometer data, processed through coordinate rotation, is first used to calculate the 10Hz Horizontal Wind Change (HWC) to reflect theinstantaneous fluctuations in horizontal wind speed(Brody-Heine, S. et al., 2023). A larger HWC

value indicates higher instability in horizontal wind speed. HWC is calculated by the difference between $u$ and $v$ at adjacent moments, as shown in Equation 1, and the Vertical Wind Change (VWC) is calculated using Equation 2.

$$\mathrm{HWC}_t = \left| \left( (u_t - u_{t-1})^2 + (v_t - v_{t-1})^2 \right)^{1/2} \right| \tag{1}$$

$$\mathrm{VWC}_t = |w_t - w_{t-1}| \tag{2}$$

To investigate the primary sources of horizontal wind speed instability, we employ Spearman correlation analysis to measure the association between the HWC series and the first-order difference series of $u$ and $v$. Spearman correlation analysis is a non-parametric statistical method used to evaluate the monotonic relationship between two variables. Unlike Pearson correlation, which is based on raw data, Spearman correlation utilizes the ranks of the variables, making it more robust to outliers and non-normally distributed data( Sedgwick, P. , 2014).

$$\begin{cases} \rho = 1 - \dfrac{6 \sum d_i^2}{n \left( n^2 - 1 \right)} \\ d_i = R(x_i) - R(y_i) \end{cases} \tag{3}$$

Among them, $\rho$ is the Spearman rank correlation coefficient, $d_i$ is the rank difference between the two variables for the $i$th pair of sample data,$R_{x_i}, R_{y_i}$ are the ranks of the $i$th sample in variables $X$ and $Y$, respectively, and $n$ is the total number of samples.



Next, we calculated the turbulent kinetic energy (TKE) using Equation 3(average period is 1 min) to measure the intensity of turbulence generated by fires under various experimental conditions ( Stull, R. B. , 2012;  Kiefer, M. et al. , 2013).

$$\text{TKE} = \left( \overline{u'^2} + \overline{v'^2} + \overline{w'^2} \right) / 2 \tag{4}$$

where $u'$, $v'$, $w'$ represent the disturbances in the three velocity components. Taking $u'$ as an example, $u' = u - \overline{u}$, $\overline{u}$ denotes the mean value of $u$ during the pre-burn period rather than the sampling period, to better reflect the mean background flow.

To observe the characteristics of turbulent heat and momentum fluxes in different burn experiments, we calculated the vertical fluxes of streamwise momentum flux $u'w'$,vertical fluxes of cross-stream momentum flux $v'w'$ and vertical kinematic heat flux $T'w'$. Additionally, we calculated the square of the friction velocity ($f_*$) as Equation 4 to quantify the magnitude of the kinematic vertical flux of horizontal momentum(Seitz, J. et al., 2024).

$$f_*^2 = \left( \overline{u'w'}^2 + \overline{v'w'}^2 \right)^{\frac{1}{2}} \tag{5}$$

## 3  Results and Discussion

### 3.1  Variations in the instability of wind

Figures 2 to 5 present the time series of 10Hz HWC and VWC derived from data recorded by 16 sonic anemometers across different experiments, covering the pre-burn, burn, and post-burn periods. To quantify the spatial heterogeneity across different columns, we calculated the overall distribution characteristics of HWC and VWC recorded in each column during the burn period, including the mean and standard deviation, as shown in Table 2 and Table 4. Table 3 and Table 5 present the Spearman correlation coefficients between the HWC and the absolute values of the first-order differences in the streamwise and cross-stream wind components of different columns during the burn period across the 4 experiments.

### 3.1.1  Variations induced by background wind level

By comparing Figures 2 and 3, it is evident that during the pre-burn period, without the interference of the burning process, both HWC and VWC increase with the average background wind speed. In the burn period, the evolutionary trends of HWC and VWC exhibit significant differences between the two experiments:the peak segments of HWC and VWC in the Burn02 experiment gradually widen from east to west, which suggests that wind speed instability persists longer as the downwind distance from the ignition boundary increases. Furthermore, it is observed that the farther a column is from the ignition boundary, the later the peak segments appears, indicating a significant time lag in the appearance of wind speed instability across the columns. In the Burn05 experiment, the peak segments of HWC and VWC at various locations typically start at the initial ignition stage and extend longer as the downwind distance from the ignition boundary increases. Unlike the Burn02 experiment, no time lag in the appearance of peak segments between different columns was observed in the Burn05 experiment. After entering the post-burn period, both HWC and VWC decreased to levels below those in the pre-burn period.

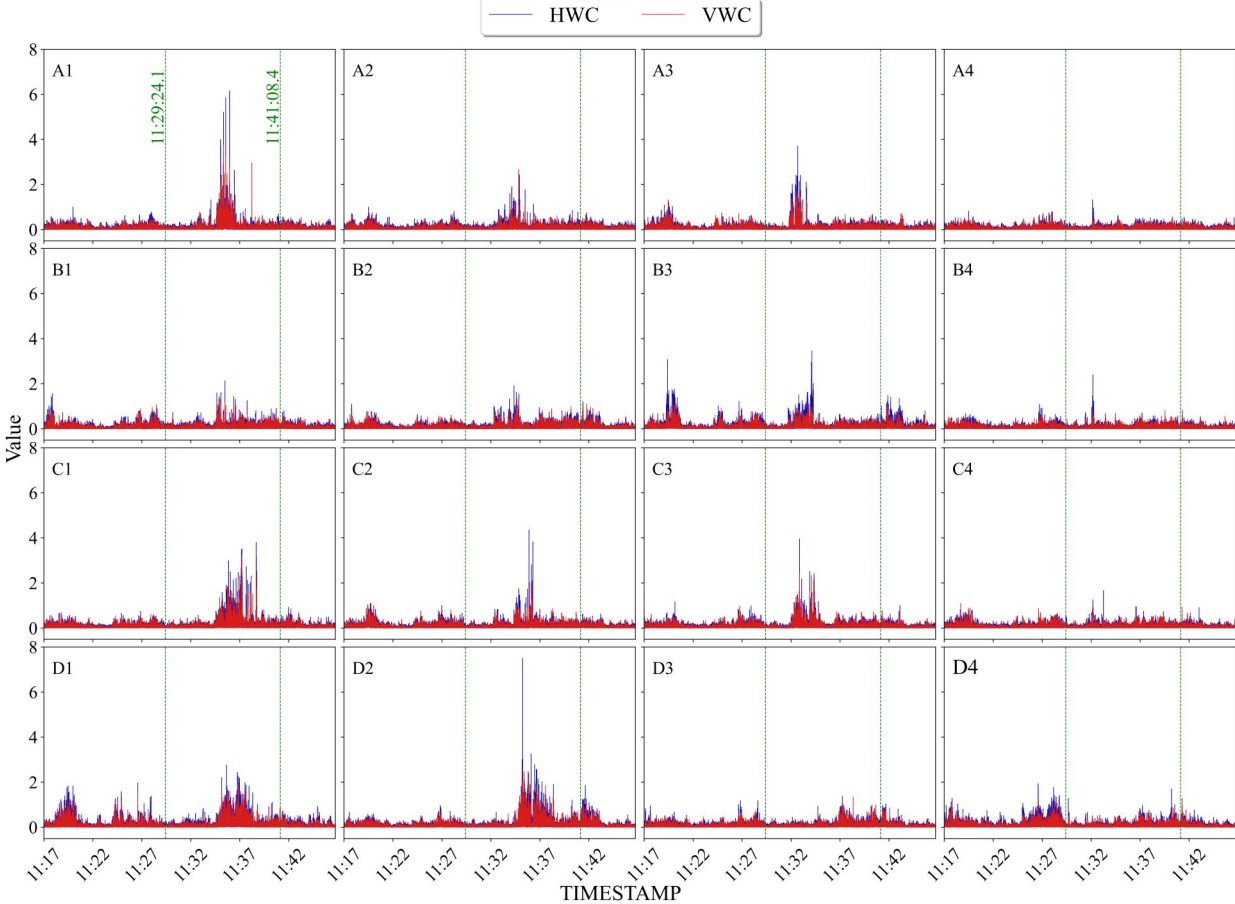

**Figure 2.** Time series of 10Hz HWC(blue) and VWC(red) in Burn02.The vertical green dashed lines indicate the start and end times of the burn period.

In the Burn02 experiment, after igniting the fire at the eastern boundary, the HWC and VWC in the A4-D4 columns showed minimal changes compared to the pre-burn period. The mean values decreased by 0.007 $m/s$ and 0.003 $m/s$, respectively, and the standard deviations decreased by 0.021 and 0.01. Starting from the A3-D3 columns, the mean values and standard deviations of HWC and VWC exceeded those of the pre-burn period and increased progressively from east to west. In the westernmost column of the burn area, the mean values of HWC and VWC were approximately 1.55 times those of the pre-burn period, with standard deviations nearly double those of the pre-burn period. In the Burn05 experiment, the column closest to the ignition boundary showed an increase in the HWC mean value by 0.029 $m/s$ compared to the pre-burn period. However, the variations in the other three indicators were consistent with those observed in the Burn02 experiment(slight variation). Moreover, the mean values and standard deviations of HWC and VWC also exhibit an increasing trend with the growing downwind distance from the ignition boundary. However, in the column farthest from the ignition boundary, both the increases

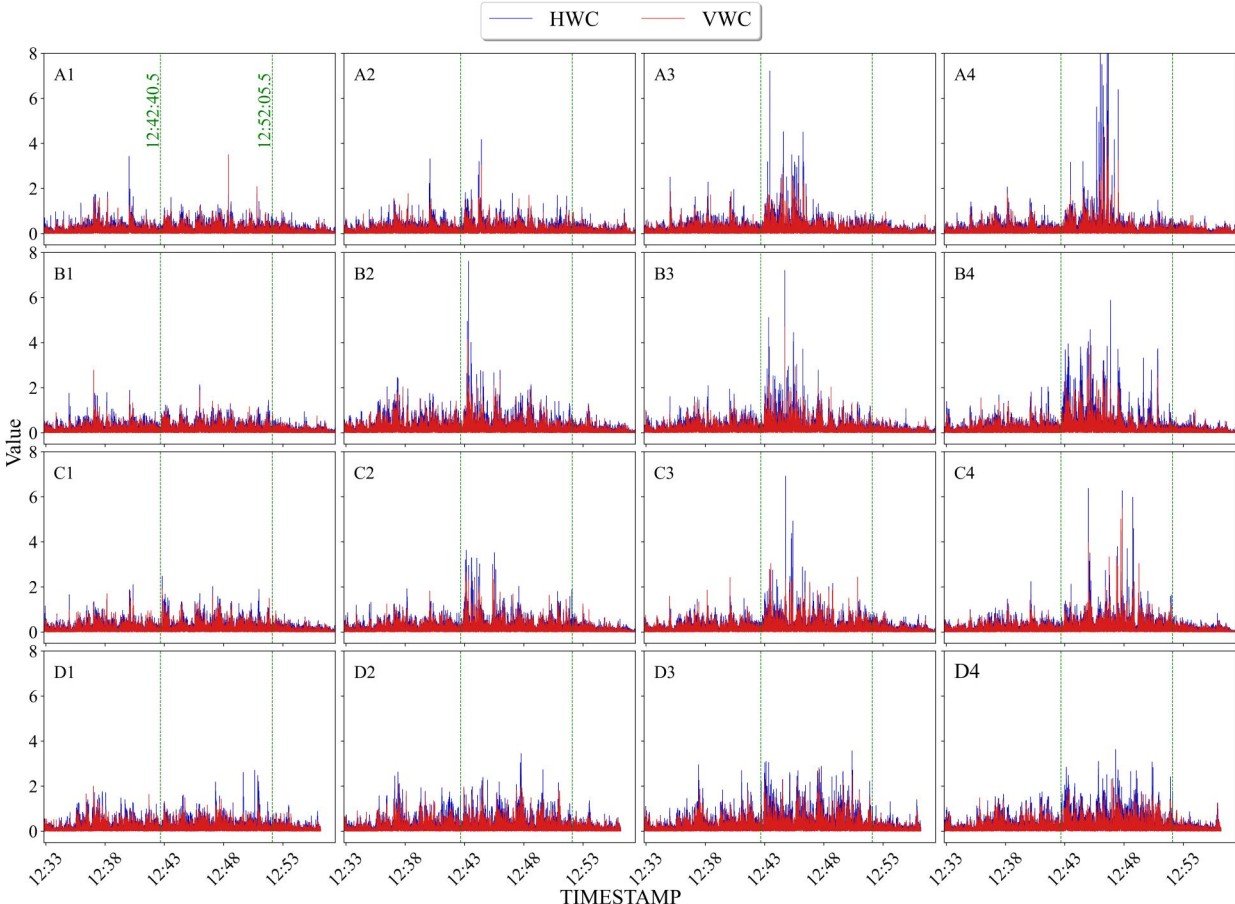

**Figure 3.** Time series of 10Hz HWC and VWC in Burn05.

in mean values and standard deviations are higher than those observed in the Burn02 experiment when compared to the pre-burn period. Upon entering the post-burn period, it was observed that in both the Burn02 and Burn05 experiments, the mean values and standard deviations of HWC and VWC decreased to levels below those of the pre-burn period. These findings are consistent with the results shown in Figures 2 and 3.

Table 3 shows that in both experiments, the correlation coefficients between HWC and the instantaneous fluctuations of the streamwise and cross-stream wind components increase with the downwind distance from the ignition boundary. The difference between the two experiments is that during the Burn02 period, with lower background wind speed, there is a stronger positive correlation between HWC and the instantaneous fluctuations of the streamwise wind speed. In contrast, during the Burn05 period, the positive correlation is stronger between HWC and the instantaneous fluctuations of the cross-stream wind component.



**Table 2.** The distribution characteristics of HWC and VWC in different burn periods under varying background wind conditions of Burn02 and Burn05.

| Item | Index | pre-burn | burn | | | | post-burn |
|------|-------|----------|------|------|------|------|-----------|
| | | Overall | A1-D1 | A2-D2 | A3-D3 | A4-D4 | Overall |
| | | | Burn02 | | | | |
| HWC | mean | 0.120 | 0.184 | 0.166 | 0.153 | 0.113 | 0.111 |
| | STD | 0.116 | 0.242 | 0.209 | 0.183 | 0.095 | 0.099 |
| VWC | mean | 0.073 | 0.113 | 0.101 | 0.092 | 0.070 | 0.066 |
| | STD | 0.086 | 0.170 | 0.139 | 0.121 | 0.076 | 0.076 |
| | | | Burn05 | | | | |
| HWC | mean | 0.229 | 0.258 | 0.321 | 0.375 | 0.394 | 0.137 |
| | STD | 0.201 | 0.205 | 0.310 | 0.374 | 0.460 | 0.108 |
| VWC | mean | 0.144 | 0.165 | 0.199 | 0.226 | 0.229 | 0.086 |
| | STD | 0.158 | 0.167 | 0.222 | 0.258 | 0.286 | 0.091 |

The distribution characteristics of HWC and VWC during the pre-burn and post-burn periods are calculated based on the data recorded by all sonic anemometers within the fire site.

**Table 3.** Results of the Spearman correlation analysis for the HWC sequences in the Burn02 and Burn05 experiments.

| | Burn02 | | | |
|------|--------|--------|--------|--------|
| Item | A1-D1 | A2-D2 | A3-D3 | A4-D4 |
| HWC vs $|\Delta u|$ | 0.773 | 0.765 | 0.767 | 0.739 |
| HWC vs $|\Delta v|$ | 0.753 | 0.741 | 0.737 | 0.712 |
| | Burn05 | | | |
| HWC vs $|\Delta u|$ | 0.707 | 0.715 | 0.726 | 0.745 |
| HWC vs $|\Delta v|$ | 0.743 | 0.770 | 0.772 | 0.769 |

$\Delta u$ denote the first-order difference of $u$.

Based on the above results, we can make the following conjectures: In low-intensity fires, wind speed instability increases closer to the fire line due to the concentration of buoyant plumes above it. This is the main reason for the lag observed in the Burn02 experiment. Higher background wind speeds typically lead to greater fire intensity, resulting in increased radiative heat and significant tilting of convection plumes. This causes substantial heat diffusion from the burned area to the unburned area, which reduces the spatial heterogeneity of HWC and VWC in the Burn05 experiment( Innocent, J. et al. , 2023;  Morvan, D. et al. , 2018).





### 3.1.2 Variations induced by fuel loading level

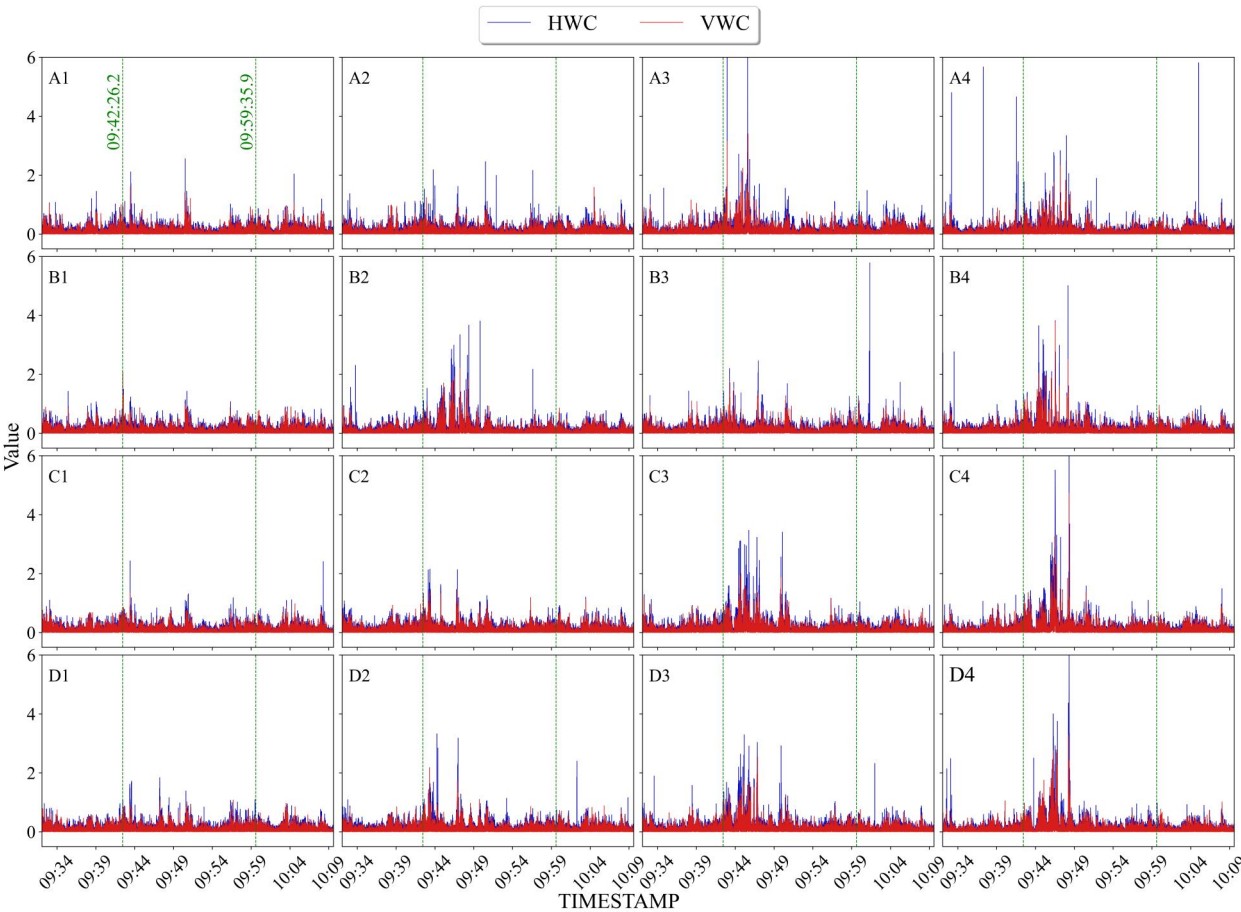

**Figure 4.** Time series of 10Hz HWC and VWC in Burn11.

By analyzing Figures 3 and 4, it was observed that in the Burn11 and Burn12 experiments, the peak segments of HWC and VWC occurred during the early phase of the burn period, with the exception of A3 in the Burn12 experiment. This indicates a lack of significant spatial heterogeneity between different columns in these two experiments. The primary distinction in the HWC and VWC time series between the two experiments was observed after entering the post-burn period. In the Burn11 experiment, both HWC and VWC returned to levels close to those of the pre-burn period. Conversely, in the Burn12 experiment,

HWC and VWC remained higher than the pre-burn period levels across most areas.

    Table 4 shows that the distribution of HWC and VWC during the pre-burn period differs between the two experiments due to the inability to maintain identical background wind conditions. During the burn period, the distribution trends of HWC and VWC are similar across different experiments, with their means and standard deviations generally increasing with the downwind distance from the ignition boundary, except for a slight decrease in columns A4-D4 compared to A3-D3 in the



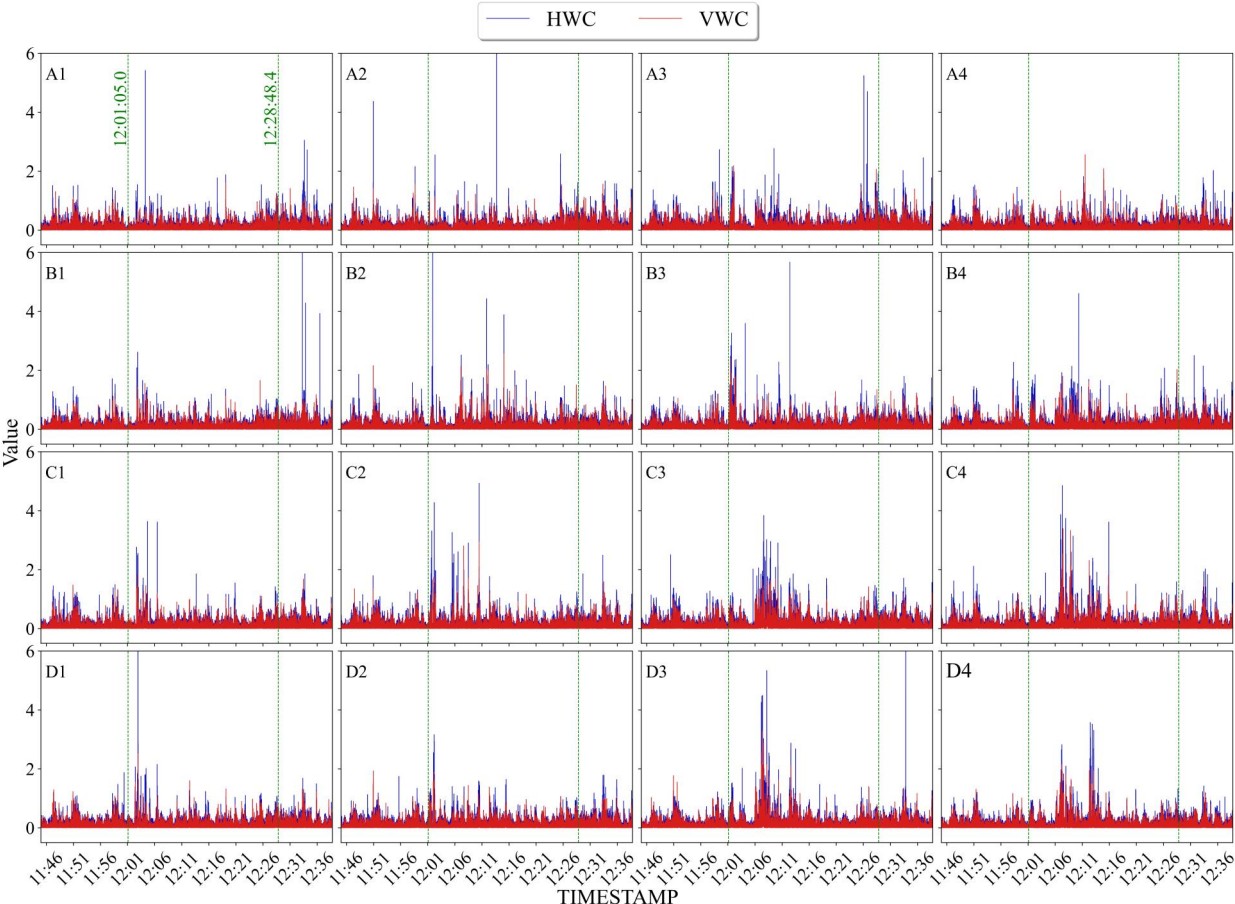

**Figure 5.** Time series of 10Hz HWC and VWC in Burn12.

Burn12 experiment. In the post-burn period of the Burn11 experiment, the means and standard deviations of HWC and VWC
are roughly equal to those in the pre-burn period. However, in the post-burn period of the Burn12 experiment, the means and
standard deviations of HWC and VWC are approximately 1.2 times higher than those in the pre-burn period. This phenomenon
is observed only in Burn12, which has the highest fuel loading among the 4 experiments.

Table 5 reveals the differences in the correlation between HWC and the instantaneous fluctuations of streamwise and cross-
stream velocities in the two experiments. In the Burn11 experiment, the correlation between HWC and both velocities increases
with the downwind distance from the ignition boundary, with the correlation to downstream velocity consistently being stronger
than to cross-stream velocity. Conversely, in the Burn12 experiment, HWC shows a stronger correlation with cross-stream
velocity fluctuations in columns A1-D1 and A2-D2.



**Table 4.** The distribution characteristics of HWC and VWC in different burn periods under varying background wind conditions (Burn11 and Burn12).

| Item | Index | pre-burn | burn | | | | post-burn |
|------|-------|----------|------|------|------|------|-----------|
| | | Overall | A1-D1 | A2-D2 | A3-D3 | A4-D4 | Overall |
| | | | | Burn11 | | | |
| HWC | mean | 0.137 | 0.144 | 0.159 | 0.179 | 0.181 | 0.134 |
| | STD | 0.125 | 0.127 | 0.168 | 0.203 | 0.233 | 0.119 |
| VWC | mean | 0.082 | 0.087 | 0.094 | 0.106 | 0.107 | 0.083 |
| | STD | 0.090 | 0.097 | 0.115 | 0.138 | 0.154 | 0.090 |
| | | | | Burn12 | | | |
| HWC | mean | 0.147 | 0.150 | 0.158 | 0.175 | 0.170 | 0.176 |
| | STD | 0.133 | 0.145 | 0.169 | 0.196 | 0.183 | 0.162 |
| VWC | mean | 0.090 | 0.090 | 0.096 | 0.104 | 0.101 | 0.107 |
| | STD | 0.101 | 0.104 | 0.0.116 | 0.129 | 0.128 | 0.112 |

**Table 5.** Results of the Spearman correlation analysis for the HWC sequences in the Burn11 and Burn12 experiments.

| | Burn11 | | | |
|------|--------|--------|--------|--------|
| Item | A1-D1 | A2-D2 | A3-D3 | A4-D4 |
| HWC vs $|\Delta u|$ | 0.741 | 0.745 | 0.750 | 0.763 |
| HWC vs $|\Delta v|$ | 0.732 | 0.740 | 0.744 | 0.748 |
| | Burn12 | | | |
| HWC vs $|\Delta u|$ | 0.732 | 0.742 | 0.752 | 0.750 |
| HWC vs $|\Delta v|$ | 0.736 | 0.745 | 0.744 | 0.747 |

In summary, for fire spread processes with high fuel loading, the effects of combustion on the wind field extend into the post-burn period. This may result from insufficient combustion of the fuel, causing smoldering to persist after the burn period ends.

## 3.2 Variations in Fire-induced turbulence intensity

The time series of TKE and its three components at each sonic anemometer location during the 4 headfire experiments are visualized in Figures 6 to 9. The distribution of TKE and its components in different periods of the 4 experiments is shown in Tables 6 to 9.



### 3.2.1 Variations induced by background wind level



**Figure 6.** Time series of 1 min averaged TKE and its three components in Burn02.The vertical green dashed lines indicate the start and end times of the burn period.

By comparing Figures 6 and 7, it is evident that during the pre-burn period, when there is no interference from the burning process, the TKE and its horizontal components are significantly higher in the Burn05 experiment, which has a higher background wind speed. However, this pattern shifts during the burn period, all of them increase substantially in the Burn02 experiment, while the increase in the Burn05 experiment is minimal. As a result, during the burn period, the TKE and its horizontal components are significantly higher in the Burn02 experiment. Additionally, during the burn period of the Burn02 experiment, the streamwise component of TKE is higher than the cross-stream component at most locations, whereas the opposite is true in the Burn05 experiment. After the burning process ends, the TKE and its horizontal components in both experiments return to levels close to those observed during the pre-burn period. The vertical component shows a consistent





**Figure 7.** Time series of 1 min averaged TKE and its three components in Burn05.

trend in both experiments: near zero during the pre-burn period, peaking significantly during the burn period, and returning to near zero during the post-burn period.

From Table 6, we observe that during the pre-burn period of the Burn02 experiment, the downstream component contributes 45.0% to the TKE, while the cross-stream component contributes 47.4%. During the burn period, the contribution of the downstream component to the TKE is higher than in the pre-burn period at all locations, and it decreases with increasing downwind distance from the ignition boundary. Conversely, the cross-stream component's contribution to TKE is lower than in the pre-burn period and increases with downwind distance. In the Burn05 experiment during the burn period, we observe the opposite trend: the downstream component's contribution to TKE is lower than in the pre-burn period at all locations and increases with downwind distance. The cross-stream component's contribution to TKE, however, is consistently higher than in the pre-burn period and gradually decreases from west to east. In the post-burn period of the Burn02 experiment, the average





**Table 6.** The distribution of TKE and its components in different periods of Burn02.

| Item | Index | pre-burn | | burn | | | post-burn |
|------|-------|----------|------|-------|-------|-------|-----------|
| | | Overall | A1-D1 | A2-D2 | A3-D3 | A4-D4 | Overall |
| TKE | mean | 0.327 | 1.378 | 1.385 | 1.453 | 1.335 | 0.447 |
| | STD | 0.175 | 0.843 | 0.808 | 0.769 | 0.781 | 0.239 |
| $u'^2/2$ | mean | 0.147 | 0.664 | 0.732 | 0.765 | 0.727 | 0.205 |
| | STD | 0.103 | 0.537 | 0.576 | 0.532 | 0.556 | 0.157 |
| $v'^2/2$ | mean | 0.155 | 0.632 | 0.595 | 0.627 | 0.576 | 0.217 |
| | STD | 0.119 | 0.363 | 0.357 | 0.397 | 0.372 | 0.218 |
| $w'^2/2$ | mean | 0.025 | 0.082 | 0.058 | 0.060 | 0.032 | 0.026 |
| | STD | 0.011 | 0.108 | 0.062 | 0.074 | 0.017 | 0.010 |

**Table 7.** The distribution of TKE and its components in different periods of Burn05

| Item | Index | pre-burn | | burn | | | post-burn |
|------|-------|----------|------|-------|-------|-------|-----------|
| | | Overall | A1-D1 | A2-D2 | A3-D3 | A4-D4 | Overall |
| TKE | mean | 0.645 | 1.051 | 1.000 | 0.991 | 1.076 | 0.695 |
| | STD | 0.274 | 0.330 | 0.312 | 0.330 | 0.367 | 0.215 |
| $u'^2/2$ | mean | 0.254 | 0.294 | 0.288 | 0.290 | 0.365 | 0.495 |
| | STD | 0.123 | 0.130 | 0.104 | 0.098 | 0.169 | 0.268 |
| $v'^2/2$ | mean | 0.316 | 0.678 | 0.618 | 0.572 | 0.564 | 0.165 |
| | STD | 0.200 | 0.264 | 0.262 | 0.256 | 0.258 | 0.094 |
| $w'^2/2$ | mean | 0.075 | 0.080 | 0.095 | 0.129 | 0.148 | 0.035 |
| | STD | 0.038 | 0.026 | 0.029 | 0.070 | 0.078 | 0.012 |

TKE was $0.447m^2/s^2$. During the same period in the Burn05 experiment, the average TKE was $0.695m^2/s^2$. Although both values are lower than those observed during the burning period, they remain higher than the pre-burn period levels.

In summary, our observations indicate that surface fires cause greater disturbances to cross-stream wind component when background wind speed is higher. This finding aligns with the results presented in Table 3. Furthermore, our preliminary analysis suggests that turbulence intensity during the burn period does not show a positive correlation with the absolute magnitude

of the background wind speed( Clements, C. B. et al. , 2015).

### 3.2.2 Variations induced by fuel loading level

From Figures 8 and 9, it is evident that during the pre-burn periods of the Burn11 and Burn12 experiments, the TKE was generally below 1 $m^2/s^2$. This suggests that the slight differences in average background wind speed between the two experiments





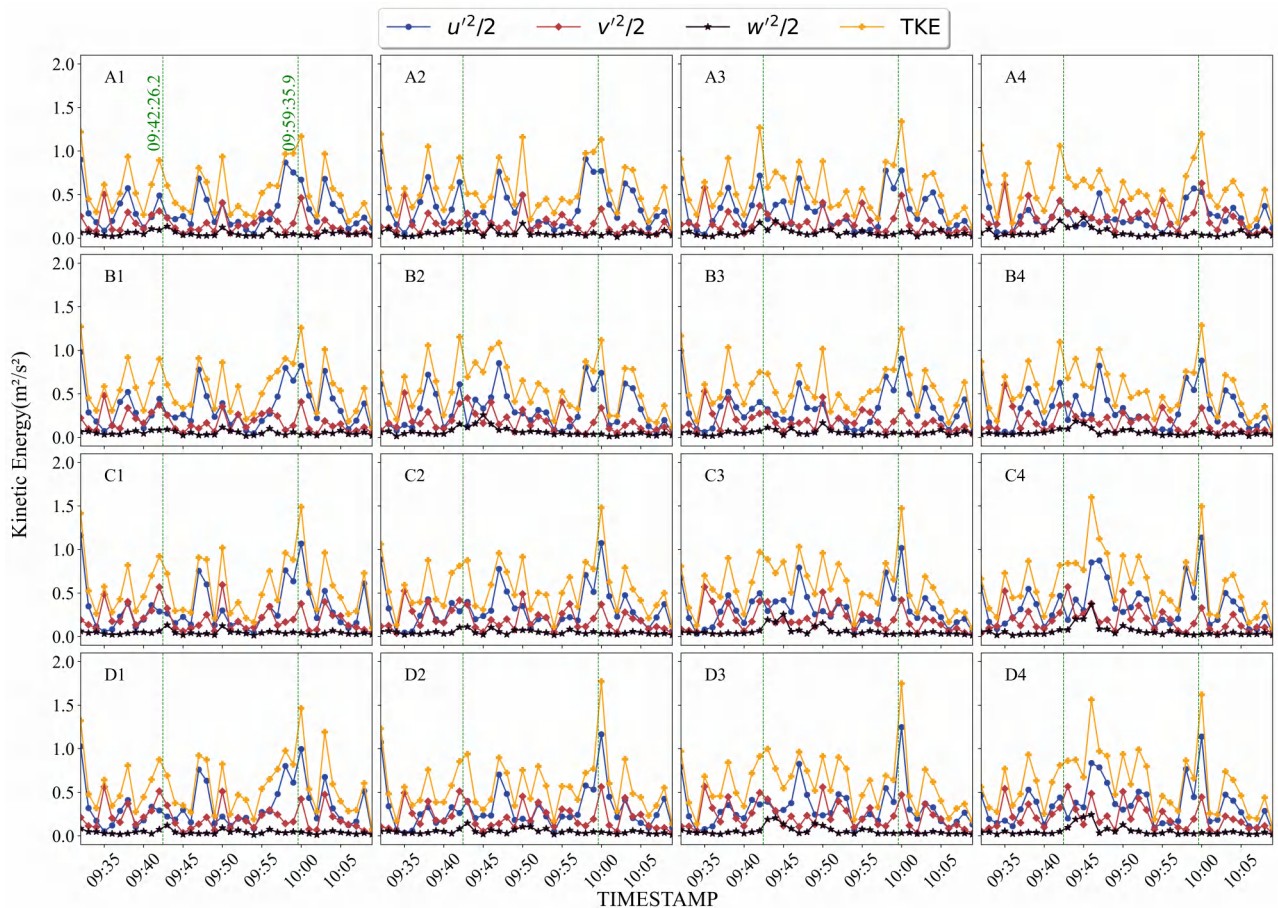

**Figure 8.** Time series of 1 min averaged TKE and its three components in Burn11.

do not significantly affect the observation of turbulence intensity variations caused by fuel loading. The differences in TKE
and its components between these two experiments are evident in two main aspects: First, during the burn period of Burn11,
the downstream component is higher than the cross-stream component, while in the Burn12 experiment, the cross-stream com-
ponent is dominant. Second, in the post-burn period, TKE and its components in the Burn11 experiment drop to levels below
those of the pre-burn period. In contrast, in the Burn12 experiment, TKE and its horizontal components remain close to the
burn period levels for a short time after the fire has extinguished.

Table 9 shows that during the pre-burn period of the Burn11 experiment, the downstream and cross-stream components
contributed $47.8\%$ and $43.4\%$ to the TKE, respectively. In contrast, during the same period in the Burn12 experiment, these
contributions were $30\%$ and $60\%$, indicating a significant difference. This discrepancy may result from an inaccurate estimation
of the ignition time in the Burn12 experiment, as these ratios align more closely with those observed during its burn period.
During the burn period of the Burn11 experiment, the downstream component consistently contributed more to TKE than the







**Figure 9.** Time series of 1 min averaged TKE and its three components in Burn12.

cross-stream component, with its contribution decreasing as the distance from the ignition boundary increased. Conversely, the cross-stream component's contribution increased with distance. However, in the same period of the Burn12 experiment, we observed the opposite trend: the rise in TKE was primarily driven by an increase in the cross-stream velocity variance. The contribution of the downstream velocity component to TKE increased with downwind distance from the ignition boundary, while the cross-stream component's contribution decreased. In the post-burn period of the Burn11 experiment, the average

TKE returned to levels close to those of the pre-burn period. However, in the same period of the Burn12 experiment, the average TKE remained relatively high at $0.877 m^2/s^2$. This indicates that in the Burn12 experiment, the impact of burning on turbulence intensity persisted into the post-burn period.

     Based on the results in Section 3.2.1 and the findings here, higher background wind speeds and increased fuel loading cause the cross-stream velocity variance to contribute significantly more to TKE than the downstream velocity variance. Both factors

are positively correlated with fire intensity. Therefore, we hypothesize that when fire intensity is below a certain threshold,



**Table 8.** The distribution of TKE and its components in different periods of Burn11

| Item | Index | pre-burn | burn | | | | post-burn |
|------|-------|----------|------|------|------|------|-----------|
| | | Overall | A1-D1 | A2-D2 | A3-D3 | A4-D4 | Overall |
| TKE | mean | 0.542 | 0.550 | 0.589 | 0.608 | 0.651 | 0.513 |
| | STD | 0.238 | 0.261 | 0.245 | 0.222 | 0.280 | 0.368 |
| $u'^2/2$ | mean | 0.259 | 0.325 | 0.328 | 0.324 | 0.344 | 0.329 |
| | STD | 0.159 | 0.226 | 0.211 | 0.187 | 0.219 | 0.261 |
| $v'^2/2$ | mean | 0.235 | 0.176 | 0.196 | 0.210 | 0.229 | 0.146 |
| | STD | 0.151 | 0.109 | 0.117 | 0.112 | 0.129 | 0.122 |
| $w'^2/2$ | mean | 0.049 | 0.050 | 0.065 | 0.074 | 0.079 | 0.039 |
| | STD | 0.028 | 0.031 | 0.042 | 0.053 | 0.067 | 0.018 |

**Table 9.** The distribution of TKE and its components in different periods of Burn12

| Item | Index | pre-burn | burn | | | | post-burn |
|------|-------|----------|------|------|------|------|-----------|
| | | Overall | A1-D1 | A2-D2 | A3-D3 | A4-D4 | Overall |
| TKE | mean | 0.584 | 0.742 | 0.741 | 0.722 | 0.733 | 0.877 |
| | STD | 0.276 | 0.345 | 0.365 | 0.354 | 0.376 | 0.403 |
| $u'^2/2$ | mean | 0.175 | 0.170 | 0.170 | 0.182 | 0.187 | 0.278 |
| | STD | 0.122 | 0.097 | 0.096 | 0.118 | 0.125 | 0.240 |
| $v'^2/2$ | mean | 0.354 | 0.516 | 0.509 | 0.471 | 0.470 | 0.528 |
| | STD | 0.185 | 0.321 | 0.344 | 0.318 | 0.326 | 0.386 |
| $w'^2/2$ | mean | 0.055 | 0.055 | 0.062 | 0.069 | 0.077 | 0.071 |
| | STD | 0.036 | 0.033 | 0.040 | 0.053 | 0.052 | 0.051 |

TKE is primarily driven by the downstream wind component. Conversely, when fire intensity exceeds this threshold, TKE is mainly influenced by the cross-stream wind component. Furthermore, in burning processes with higher fuel loading, the impact of the fire on TKE extends into the post-burn period, likely due to the ignition-induced fire behavior( Clark, K. L. et al. , 2015; Arreola Amaya, M. et al., 2020).

### 3.3 Variations in fire-induced shear stress

The time series of friction velocity squared, streamwise kinematic momentum flux, and cross-stream kinematic momentum flux at various locations during the three periods of the 4 experiments are shown in Figures 10 to 13. The mean and standard deviation of momentum flux and $f_*^2$ at different locations during the pre-burn, burn, and post-burn periods in the 4 experiments are shown in Tables 10 to 13.





### 3.3.1 Variations induced by background wind level



**Figure 10.** time series of friction velocity squared, streamwise kinematic momentum flux, and cross-stream kinematic momentum flux at various locations during the three periods of Burn02.

By comparing Figures 10 and 11, we observed that during the pre-burn period without fire interference, an increase in background wind speed resulted in larger fluctuations in $f_*^2$, $u'w'$, and $v'w'$. When the Burn02 experiment entered the burn period, the amplitude of changes in $f_*^2$, $u'w'$, and $v'w'$ increased with distance from the ignition boundary, a trend not observed in the Burn05 experiment. Another significant difference between the two experiments during the burn period is that in Burn02,

the blue and red curves alternately lead at most locations, while in Burn05, the pattern from the pre-burn period remains: the red curve is always positive, and the blue curve is always negative. In the post-burn period of the Burn02 experiment, $f_*^2$, $u'w'$, and $v'w'$ returned to levels close to the pre-burn period, whereas in the Burn05 experiment, f at some locations was higher than during the burn period, and uw was more negative, as observed at location A4.



**Figure 11.** time series of friction velocity squared, streamwise kinematic momentum flux, and cross-stream kinematic momentum flux at various locations during the three periods of Burn05.

**Table 10.** The distribution of $f_*^2$, $u'w'$, and $v'w'$ in different periods of Burn02.

| Item | Index | pre-burn | burn | | | | post-burn |
|------|-------|----------|------|------|------|------|-----------|
| | | Overall | A1-D1 | A2-D2 | A3-D3 | A4-D4 | Overall |
| $u'w'$ | mean | -0.032 | -0.113 | -0.044 | 0.003 | 0.018 | 0.001 |
| | STD | 0.041 | 0.303 | 0.209 | 0.197 | 0.133 | 0.066 |
| $v'w'$ | mean | 0.000 | 0.051 | -0.002 | -0.027 | -0.050 | -0.024 |
| | STD | 0.046 | 0.218 | 0.143 | 0.164 | 0.103 | 0.051 |
| $f_*^2$ | mean | 0.054 | 0.244 | 0.181 | 0.209 | 0.142 | 0.069 |
| | STD | 0.044 | 0.309 | 0.183 | 0.152 | 0.105 | 0.052 |





**Table 11.** The distribution of $f_*^2$, $u'w'$, and $v'w'$ in different periods of Burn05.

| Item | Index | pre-burn | burn | | | | post-burn |
|------|-------|----------|------|------|------|------|-----------|
| | | Overall | A1-D1 | A2-D2 | A3-D3 | A4-D4 | Overall |
| $u'w'$ | mean | -0.057 | -0.064 | -0.065 | -0.076 | -0.088 | -0.082 |
| | STD | 0.076 | 0.055 | 0.058 | 0.076 | 0.081 | 0.099 |
| $v'w'$ | mean | 0.064 | 0.053 | 0.085 | 0.081 | 0.056 | -0.008 |
| | STD | 0.065 | 0.094 | 0.081 | 0.097 | 0.107 | 0.038 |
| $f_*^2$ | mean | 0.106 | 0.123 | 0.131 | 0.144 | 0.150 | 0.107 |
| | STD | 0.079 | 0.060 | 0.065 | 0.082 | 0.080 | 0.081 |

Tables 10 and 11 indicate that during the pre-burn period of the Burn02 experiment, the mean value of $f_*^2$ was $0.054 m^2 s^{-2}$.

As the experiment progressed into the burn period, $f_*^2$ increased progressively with distance from the ignition boundary, peaking at $0.244 m^2 s^{-2}$. In the post-burn period, the mean value of $f_*^2$ decreased to $0.069 m^2 s^{-2}$, falling between the pre-burn and burn periods. In contrast, the Burn05 experiment showed a similar trend, but the peak value of $f_*^2$ was only 1.4 times that of the pre-burn period, significantly lower than the fourfold increase observed in Burn02. During the burn period of the Burn02 experiment, $u'w'$ exhibited the most significant change near the ignition boundary, shifting from $-0.032 m^2 s^{-2}$

to $0.018 m^2 s^{-2}$. However, as the distance from the ignition boundary increased, $u'w'$ gradually returned to pre-burn values. In the Burn05 experiment, it became increasingly negative with greater distance from the ignition boundary. For $v'w'$ in the Burn02 experiment, the mean value near the ignition boundary was $-0.05 m^2 s^{-2}$. As the distance increased, vw turned positive, reaching $0.051 m^2 s^{-2}$. Conversely, in the Burn05 experiment, it remained positive throughout.

The results suggest that there is no positive correlation between background wind speed and the magnitude of $f_*^2$. Addition-

310 ally, the influence of background wind speed on the temporal and spatial trends of $u'w'$ and $v'w'$ requires further investigation( Kiefer, M. T. et al. , 2014).

### 3.3.2 Variations induced by fuel loading level

Figures 12 and 13 first illustrate a significant difference: in the Burn11 experiment, the red line leads the blue line in all periods, while in the Burn12 experiment, the blue line consistently leads,which is related to the environmental conditions during the

315 experiments and not to the fuel loading we are studying. Secondly, it is evident that in the post-burn period of the Burn11 experiment, the three variables $f_*^2$, $u'w'$, and $v'w'$ return to levels comparable to the pre-burn period. However, in the same period of the Burn12 experiment, these variables at most locations are close to or even exceed the levels observed during the burn period.

Tables 12 and 13 indicate that the mean values of $f^2$ during the pre-burn periods of the two experiments were $0.095\ m^2 s^{-2}$

and $0.097\ m^2 s^{-2}$, respectively. In both experiments, the maximum values of $f^2$ during the burn periods were observed in the column farthest from the ignition boundary, with these maximum values being approximately 1.2 times those of the pre-burn





**Figure 12.** time series of friction velocity squared, streamwise kinematic momentum flux, and cross-stream kinematic momentum flux at various locations during the three periods of Burn11.

**Table 12.** The distribution of $f_*^2$, $u'w'$, and $v'w'$ in different periods of Burn11.

| Item | Index | pre-burn | burn | | | | post-burn |
|------|-------|----------|------|------|------|------|-----------|
| | | Overall | A1-D1 | A2-D2 | A3-D3 | A4-D4 | Overall |
| $u'w'$ | mean | 0.016 | 0.017 | 0.003 | 0.009 | 0.030 | 0.014 |
| | STD | 0.069 | 0.067 | 0.085 | 0.095 | 0.119 | 0.075 |
| $v'w'$ | mean | 0.062 | 0.046 | 0.060 | 0.051 | 0.042 | 0.030 |
| | STD | 0.064 | 0.064 | 0.069 | 0.075 | 0.067 | 0.039 |
| $f_*^2$ | mean | 0.095 | 0.085 | 0.103 | 0.115 | 0.113 | 0.070 |
| | STD | 0.063 | 0.060 | 0.071 | 0.065 | 0.092 | 0.058 |





**Figure 13.** time series of friction velocity squared, streamwise kinematic momentum flux, and cross-stream kinematic momentum flux at various locations during the three periods of Burn12.

**Table 13.** The distribution of $f_*^2$, $u'w'$, and $v'w'$ in different periods of Burn12.

| Item | Index | pre-burn | burn | | | | post-burn |
|------|-------|----------|------|------|------|------|-----------|
| | | Overall | A1-D1 | A2-D2 | A3-D3 | A4-D4 | Overall |
| $u'w'$ | mean | 0.061 | 0.043 | 0.037 | 0.042 | 0.050 | 0.088 |
| | STD | 0.073 | 0.050 | 0.052 | 0.070 | 0.082 | 0.138 |
| $v'w'$ | mean | 0.004 | 0.027 | 0.006 | 0.002 | -0.006 | -0.019 |
| | STD | 0.082 | 0.090 | 0.100 | 0.101 | 0.120 | 0.106 |
| $f_*^2$ | mean | 0.097 | 0.094 | 0.090 | 0.098 | 0.120 | 0.137 |
| | STD | 0.080 | 0.066 | 0.078 | 0.085 | 0.096 | 0.140 |





periods. In the post-burn period of the Burn11 experiment, the mean and standard deviation of $f^2$ were lower than during the pre-burn period. Conversely, in the same period of the Burn12 experiment, the mean and standard deviation of $f^2$ were $0.137 m^2 s^{-2}$ and 0.140, respectively, exceeding the maximum values observed during the burn period.

In summary, fuel loading significantly impacts momentum flux and $f^2$ by enhancing momentum transfer during the post-burn period in areas with higher vegetation levels( Prohanov, S. A. et al. , 2017).

### 3.4 Variations in fire-induced turbulent heat flux

Figures 14 to 17 show the time series of 1 min average turbulent heat flux at each sonic anemometer position in the four experiments. The mean and standard deviation of turbulent heat flux during each period of the four experiments are shown in Tables 14 to 15.

#### 3.4.1 Variations induced by background wind level

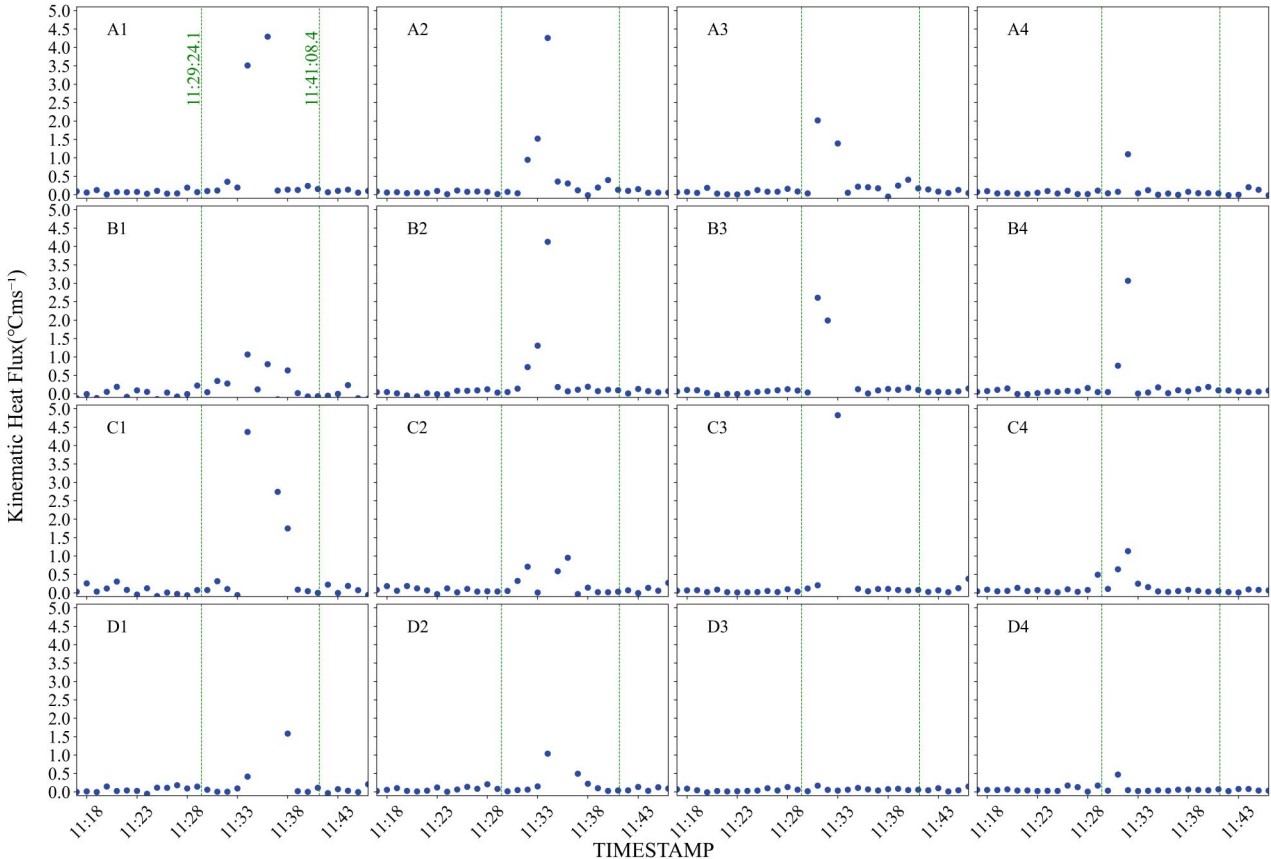

**Figure 14.** Time series of 1 min average turbulent heat flux at each sonic anemometer position during the three periods of Burn02.



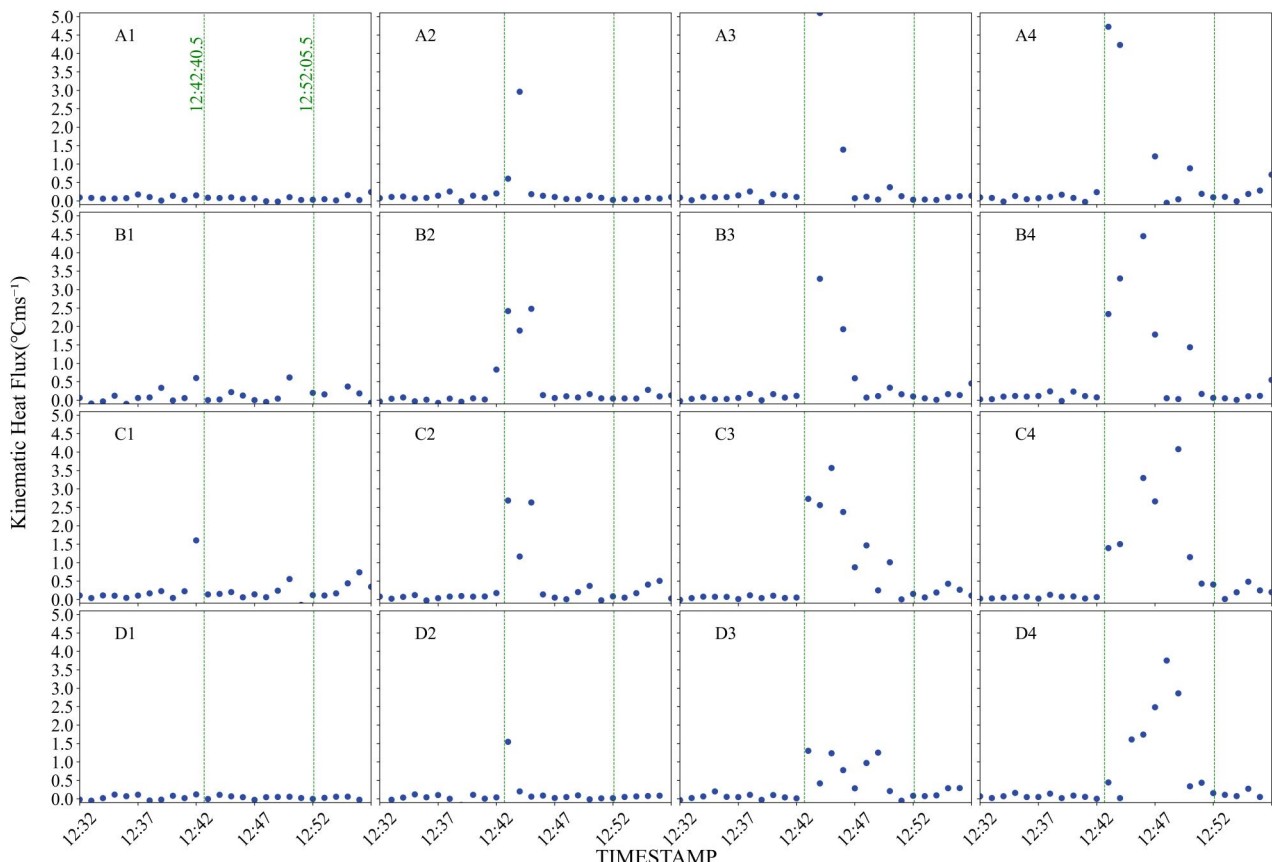

**Figure 15.** Time series of 1 min average turbulent heat flux at each sonic anemometer position during the three periods of Burn05.

Figures 14 and 15 show that during the burn periods of the Burn02 and Burn05 experiments, the further from the ignition boundary, the longer the duration of the heat flux increase. This trend is more pronounced in Burn05, where the heat flux in the easternmost column remained high throughout the burn period, while in columns A2-D2, the increase in heat flux was

335 only observed in the first third of the burn period. This phenomenon may be attributed to two factors: first, higher wind speeds lead to greater fire intensity, resulting in increased radiative heat transfer from the ignition area to the unburned area; second, higher background wind speeds cause more intense heat diffusion. By analyzing Table 14, we identified a common trend in the Burn02 and Burn05 experiments: during the burn period, turbulent heat flux generally increased with distance from the ignition boundary, except for columns A3-D3 in Burn02. The maximum values were observed in the columns farthest from the

340 ignition boundary. In the post-burn period, turbulent heat flux quickly decreased to levels slightly above the pre-burn values.

In summary, the variations in heat flux due to background wind speed are evident in two aspectshigher background wind speed will cause the growth of heat flux to last longer; moreover in combustion with higher background wind speed, the multiple increase in heat flux compared to the pre-burn period is higher.



**Table 14.** Distribution of heat flux at different periods in Burn02 and Burn05.

| Index | pre-burn | | burn | | | post-burn |
|---|---|---|---|---|---|---|
| | | | Burn02 | | | |
| mean | 0.052 | 1.876 | 0.934 | 1.175 | 0.204 | 0.071 |
| STD | 0.061 | 3.804 | 2.200 | 2.701 | 0.487 | 0.079 |
| | | | Burn05 | | | |
| mean | 0.091 | 0.085 | 0.530 | 1.588 | 2.169 | 0.158 |
| STD | 0.158 | 0.143 | 0.892 | 2.187 | 2.199 | 0.168 |

### 3.4.2 Variations induced by fuel loading level

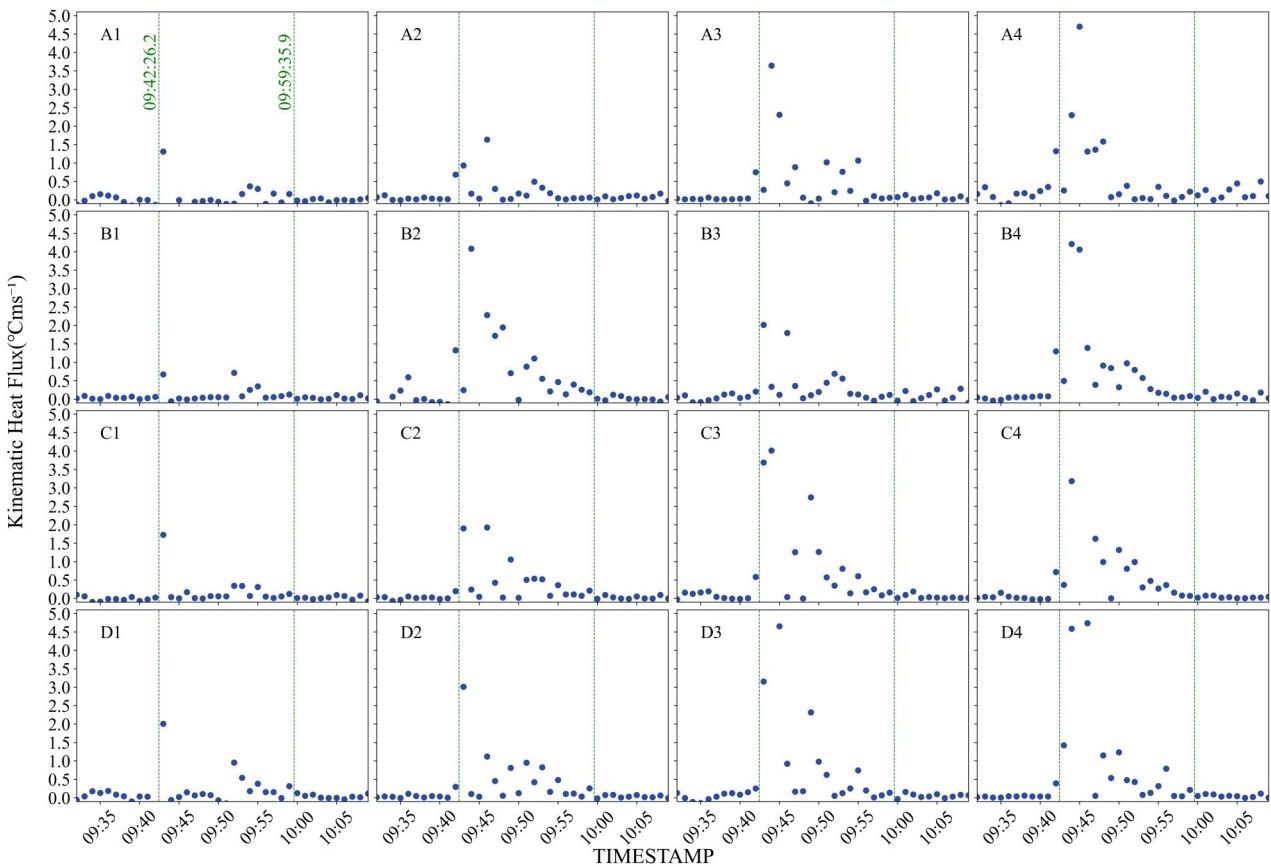

**Figure 16.** Time series of 1 min average turbulent heat flux at each sonic anemometer position during the three periods of Burn11.



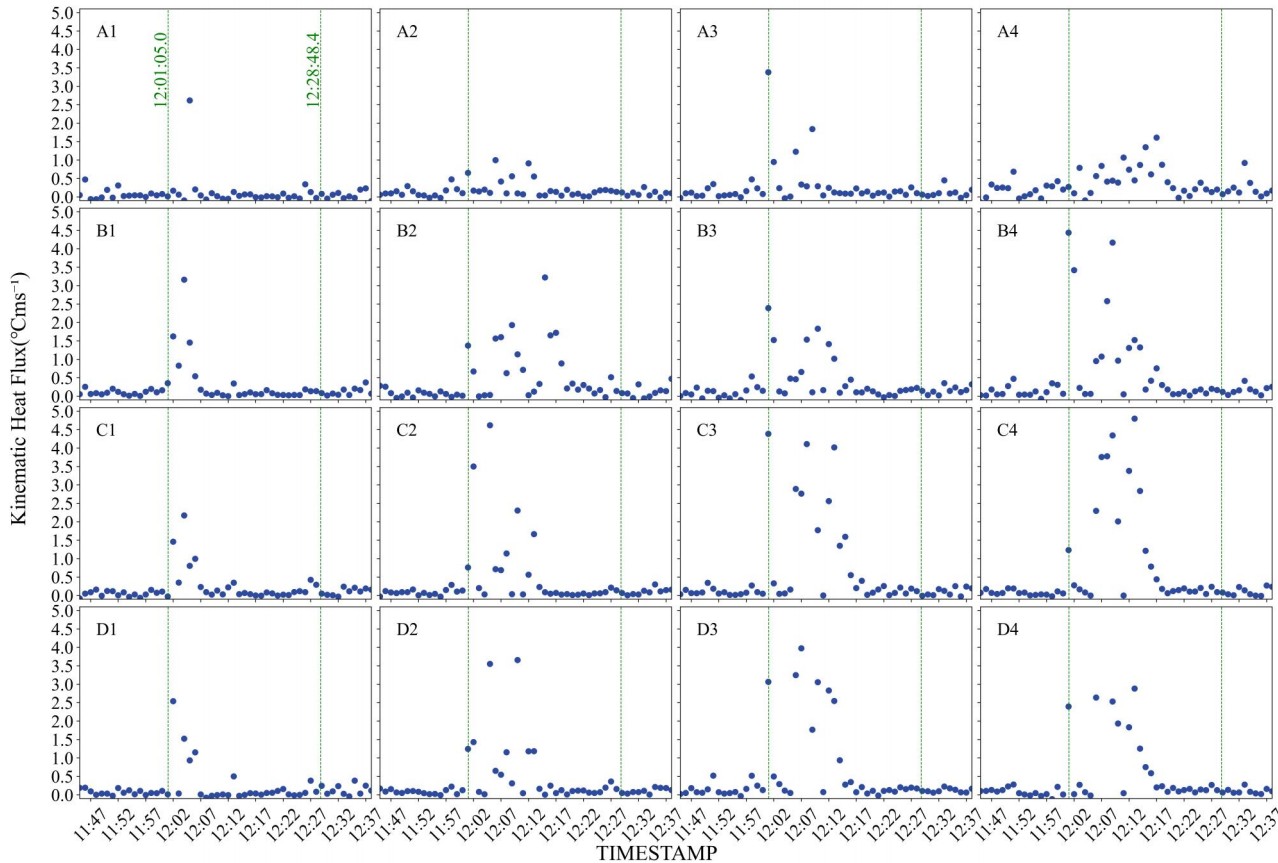

**Figure 17.** Time series of 1 min average turbulent heat flux at each sonic anemometer position during the three periods of Burn12.

By comparing Figures 16 and 17, we can see that during the burn period, the heat flux closest to the ignition boundary increased more significantly in the Burn12 experiment. In both experiments, the other three columns followed a similar trend: the further from the ignition boundary, the greater the increase in heat flux. This increase mainly occurred in the early and middle stages of the burn period, with levels returning to near pre-burn cycle values in the latter half. The differences in the duration of heat flux increase observed in the first two experiments were not present in the Burn11 and Burn12.

Table 15 shows that in the Burn11 and Burn12 experiments, both the mean and standard deviation of heat flux increased with the distance from the ignition boundary. In the post-burn period of the Burn11 experiment, the mean heat flux dropped to $0.051°\mathrm{C}ms^{-1}$, about 5% of the maximum value during the burn period. In the Burn12 experiment, the mean heat flux during the same period decreased to $0.121°\mathrm{C}ms^{-1}$, roughly 15% of the maximum value. This suggests that higher fuel loading results in a less pronounced reduction in heat flux after the flames are extinguished, likely due to continued ignition and smoldering

effects( Mueller, E. V. et al. , 2017).





**Table 15.** Distribution of heat flux at different periods in Burn11 and Burn12.

| Index | pre-burn | burn | | | | post-burn |
|-------|----------|------|---|---|---|-----------|
| | | Burn11 | | | | |
| mean | 0.084 | 0.180 | 0.653 | 0.928 | 1.099 | 0.051 |
| STD | 0.222 | 0.387 | 1.125 | 1.533 | 1.618 | 0.081 |
| | | Burn12 | | | | |
| mean | 0.092 | 0.268 | 0.524 | 0.714 | 0.872 | 0.121 |
| STD | 0.111 | 0.577 | 0.861 | 1.240 | 1.306 | 0.123 |

## 4    Conclusions

In the Silas Little Experimental Forest in New Jersey, USA, 35 low-intensity fire experiments were conducted on $10m{\times}10m$ plots, and high-density measurements of turbulence were taken. This laid an important foundation for the study of finer-scale turbulence in fire environments. From these, 4 downwind fires were selected, forming two sets of control experiments to investigate the impact of background average wind speed and fuel load levels on the spatiotemporal characteristics of fire-induced turbulence. The temporal characteristics mainly refer to the differences between the periods before, during, and after the fire, whereas the spatial characteristics primarily denote the trend of turbulence characteristics with respect to the distance from the ignition boundary during the combustion period.

The comparison using HWC and VWC as dependent variables indicates that when the background wind speed is low, the fire site wind speed response is primarily concentrated at the fireline location due to a significant lag in peak segments occurrence time at varying distances from the ignition boundary. However, this time lag is not observed when the background wind speed is high. In such cases, the response of instantaneous wind speed changes at various fire site locations begins at the start of the burn period and persists for some time. This may result from the significant tilt of the convective plume under high background wind speed, causing substantial heat to spread from the combustion zone to the unburned zone. The impact of fuel load changes on turbulence characteristics is primarily seen in how higher fuel loads extend the fire site wind speed response into the post-burn period. This may be due to the propensity for incomplete combustion under these conditions, resulting in strong smolder phenomena during the post-burn period. Spearman correlation analysis of the first-order differences between HWC and instantaneous changes in downstream and crosswind wind speeds indicates that both background wind speed and its increase will lead to HWC being more influenced by the cross-stream component. The comparison using TKE as dependent variables indicates that the increase in background wind speed and fuel load did not result in a significant change in the magnitude of turbulence intensity. In the four experiments, TKE consistently remained below $3m^2s^{-2}$, indicating that no substantial turbulent environment was formed above the fire site during the fires. Notably, higher background wind speed and fuel load resulted in the variance of the crosswind velocity component contributing significantly more to TKE than the variance of the downstream velocity component. Since both variables are related to fire intensity, it is hypothesized that when fire intensity



is below a certain threshold, TKE is mainly driven by the downstream wind component; when fire intensity exceeds this threshold, the dominant component of TKE shifts to the crosswind component. The comparison using turbulent momentum flux and shear stress indicates that the increase in background wind speed weakened the trend of turbulent momentum flux and shear stress fluctuations with the increasing distance from the smoldering boundary. Under high background wind speed conditions, high downstream momentum air is consistently transferred downward, while under low background wind speed

conditions, this direction is not fixed. The impact of fuel load on momentum transfer is mainly observed when the fuel load is high, as the momentum flux and shear stress during the post-burn period do not significantly differ from those during the burn period. The comparison using heat flux indicates that as the background wind speed increases, the duration of heat flux growth during the burn period is prolonged. The higher the fuel load, the smaller the difference between the heat flux levels during the post-burn period and the burn period.

In the future, fire behaviors (such as spread rate and fire intensity) will be parameterized based on the data from this paper, as well as infrared image data and thermocouple data from various experiments, to analyze their impact on fire site turbulence characteristics. Additionally, quadrant analysis and spectral and co-spectral analyses will be conducted for more in-depth research.

*Code and data availability.*   All-data-generated-during-the-conduct-of-the-study can be found at https://github.com/lcy981009/All-data-generated-

during-the-conduct-of-the-study.git. All Code used for caculating and ploting during the study can be found at https://github.com/lcy981009/Code-used-for-caculating-and-ploting.

*Author contributions.*   Chuanying Li:Writing-original-draft, Methodology. Xingdong Li:Data analysis and visualization. Honhgyang Zhao:Manuscript revision and typesetting. Dandan Li:Validation.

*Competing interests.*   The authors declare that they have no known competing financial interests or personal relationships that could have

appeared to influence the work reported in this paper.

*Acknowledgements.*   We extend our gratitude to the project "Multi-scale Analyses of Wildland Fire Combustion Processes in Open-canopied Forests using Coupled and Iteratively Informed Laboratory-, Field-, and Model-based Approaches (RC-2641)" for providing the data.





**Financial support**

This work was supported by National Key Research and Development Program of China (Grant No.2023YFC3006900); Cen-
tral Finance Forestry Science and Technology Promotion Demonstration Project(Grant No.Hei[2024]TG11);the Open Research
Fund of Anhui Province Key Laboratory of Machine Vision Inspection (Grant No.KLMVI-2023-HIT-06).



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
