# Peer review of "Variations in Turbulence Characteristics of Head Fires Induced by Background Wind and Fuel Loading Levels"

_EGUsphere, 2024_

## Referee Comment (RC2)

This manuscript presents the analysis of spatiotemporal fire-induced turbulence characteristics under varying ambient wind speeds and fuel loading. The fire-induced turbulence was characterized in terms of variances, TKE, sensible heat flux, and momentum fluxes along with their mean values. The field experiments are unique in a way that turbulence generated by low intensity head fires was measured by a sense network of the sonic anemometers above the ground. The small burn plots (10 m x 10 m) allow for more controlled fire spread and behavior to be observed and analyzed.

On the downside, the analysis of fire behavior presented is a bit too speculative and justifications of the interpretation are generally weak. My biggest concern about this manuscript is the lack of fire behavior analysis. I outlined my suggestions below to improve the manuscript. The dataset presented have potential to improve our understanding of fire-atmosphere interactions in very fine scale. However, I do not think the manuscript has an original contribution to wildfire research community to advance their/our current knowledge at this stage. The analysis needs to be a bit more in depth, and there appears to be data (IR camera, radiometers) to implement that.

 Major comments

Time series of background wind speeds for each burn should be shown in addition to the mean values presented in Table 1 to make sure the background wind conditions did not change significantly during the fire. If it did, then the assumption of low vs. high wind case falls apart. This can be done by plotting wind speeds measured at the control tower before, during, and after the fire all together from the 4 burn experiments. I would suggest plotting 4 lines in 1 plot with different color.

In my opinion, showing fire spread across the 4x4 sensors measured by the IR camera is critical in terms of interpreting fire-induced turbulence in time and space. Fireline intensity and shape is rarely homogeneous or uniform in space and time in nature, and observed turbulence should be strongly affected in both horizontal and vertical directions. I assume that higher fuel loading does not necessarily mean higher fireline intensity or fuel/flame temperature. I suggest to create a plot with 3 panels for each burn to show the IR camera temperatures at 3 selected time stamps (so total of 12 plots). See Fig. 3 in Seitz et al. (2024) or Figs 2 and 5 in Katurji et al. (2021) as examples. The IR camera can show where the fire front was, by what time the fire reached the individual sonic anemometers, and how hot smoldering was. Also, make sure to show the location of 12 sonic anemometers in the plot.

Line 261: The crosswise variance was already dominant before the presence of fire during Burn 12, and the fire did not change this dominant flow/turbulence behavior as Fig. 9 and Table 9 show. Therefore, the sentence 'First, during the burn period of Burn 11, … while in the Burn 12 experiment the cross-stream component is dominant.' Is misleading and not accurate. It simply means that the ambient flow in the crosswise velocity direction was more turbulent in the area due to canopy structure for example.

Line 267: inaccurate estimation of the ignition time in Burn 12 should be corrected using the in-situ radiometer data. The data should be able to show when the fireline started crossing beneath the sonic anemometers. Alternatively, you can use the IR camera data to adjust the ignition time estimate.

Line 278: Figure 9 and Table 9 do not indicate that 'higher background wind speeds and/or increased fuel loading caused the cross-stream velocity variance to contribute significantly to the TKE than streamwise velocity variance. The results indicate that the fire did not modify the fact that $v$-velocity variance dominated the TKE under stronger background wind speed and higher fuel loading.

Line 280: I do not see how the fire intensity was measured or calculation. This is very important as the strength of fire-induced turbulence depends heavily on the fire/fireline intensity (as well as the degree of coupling between fire and atmosphere).

Line 281: The authors state that 'when fire intensity exceeds this threshold, TKE is mainly influence by the cross-stream wind component.' Again, I think this is not a correct interpretation of your result. See my previous comment (Line 261).

Line 283: It is more likely that higher fuel loading resulted in relatively high heat release during the smoldering phase. The authors can easily show this by analyzing radiometer and IR camera data. I am not convinced that it is due to the ignition-induced fire behavior unless the authors can show evidence of fire behavior at the time of ignition.

Results in general: Please compare your findings with Seitz et al. (2024) and discuss the similarities and contrasts between them because the analysis of this manuscript is essentially very similar but using different experimental data (i.e. different weather and fuel conditions) from the same field campaign. That will make this paper to contribute more to understanding of combustion processes and fire-atmosphere interactions.

Line 368: The authors state that 'This may result from the significant tilt of the convective plume under high background wind speed, …' Would it be possible to prove it by photo images (if available) or using thermocouple data? That would make the statement much more convincing.

Line 371: smoldering phenomena during the post-burn period can be easily quantified using the IR camera (and radiometer) data instead of saying 'This may be due to ….'

Line 383: The increasing distance from the smoldering boundary could be true, but there is no evidence of smoldering behavior (how hot, and spatial variability) is shown in this paper. This can be shown using the IR camera data. If the smoldering phase has strong impact on atmospheric turbulence, then it should be shown explicitly.

Line 390: How can we parametrize fire behavior using your data? It would be useful for fire behavior modelers to know this. Has anyone done that using experimental data in the past? If so, how?

Minor comments

Please make sure that the manuscript is correctly formatted. I see lots of missing space before '( ' for example.

Also, I would suggest the terms 'streamwise velocity' to indicate the u-velocity that is rotated into the streamwise direction and 'crosswise velocity' to indicate the v-velocity. Make sure to keep the wording consistent throughout the manuscript.

Table1: add fuel moisture content if known. It affects fire intensity. I noticed that the RH is much lower during Burn 2.

References

Katurji, M., Zhang, J., Satinsky, A., McNair, H., Schumacher, B., Strand, T., et al. (2021). Turbulent thermal image velocimetry at the immediate fire and atmospheric interface. Journal of Geophysical Research: Atmospheres, 126, e2021JD035393. https://doi.org/10.1029/2021JD035393

Seitz, Joseph, et al. "Atmospheric turbulence observed during a fuel-bed-scale low-intensity surface fire." Atmospheric Chemistry and Physics 24.2 (2024): 1119-1142.

---

## Author Comment (AC1)

We sincerely appreciate you for your thoughtful and insightful comments on our manuscript. Your constructive feedback has played a crucial role in refining and strengthening the quality of our work. We highly appreciate the time and effort your dedicated to providing valuable suggestions, which have undoubtedly enhanced the quality of our article.

Below, we present our response to each of the comments.

*This manuscript presents findings from four surface fire experiments on small plots (10m x 10m), aimed at documenting the effects of background wind speed and fuel loading on turbulence. Although the paper is generally well-written and the figures are clearly presented, there are significant issues that undermine its credibility, as outlined below.*

**Response:** First of all, We appreciate your kind words regarding the clarity of the writing and the the quality of the figures.

We would like to emphasize that we have consistently upheld a rigorous approach throughout our scientific research. We sincerely apologize for any concerns about the paper's credibility that were caused by our lack of rigor. Rest assured, we have taken your feedback seriously and have thoroughly addressed these issues in the revision process to further enhance the rigor and credibility of the paper.

*1. Incorrect Citations: The manuscript inaccurately cites Ottmar et al. (2015) for field experiments that were actually conducted in 2018-2019. This discrepancy in citation undermines the clarity and accuracy of the research. Correct citation is critical to maintain scientific integrity.*

**Response:** Thank you for pointing out the inaccurate citation in line 110 of the original manuscript. We have identified that this error was caused by a technical oversight while using BibTeX for referencing. The correct citation should have been Gallagher et al. (2023), but due to an inadvertent mistake, it was misattributed to Ottmar et al. (2015). We have thoroughly reviewed all citations and corrected this error during the revision process to ensure the accuracy of the references.

*2. Misuse of Figures: A figure from a previously published paper is reproduced and labelled as "adapted" although there is no indication that the figure has been modified. This could be seen as a direct reproduction without proper attribution, which raises concerns regarding proper credit.*

**Response:** We acknowledge that there was an oversight in this matter. The original figure was not modified, yet it was labeled as "adapted". We have recognized this issue and have already started the process of requesting permission from the original author for the use of the figure. Once permission is granted, we will mark the figure as

"reproduced with permission" in the revised version, or apply any other appropriate attribution as per the terms of the permission, ensuring compliance with copyright and citation guidelines.

*3. Misrepresentation of Data:The manuscript presents the data as relating to wildfire behaviour, implying that it contributes to understanding fire-induced weather. In reality, the data comes from low-intensity prescribed fires, which were designed to study specific fire phenomena in controlled environments. The description in the introduction, which frames the research as focusing on "Turbulence from wildfires," is somewhat misleading given the smaller scale and different context of the prescribed fires.*

**Response:** We appreciate your valuable feedback on our manuscript and acknowledge that data derived from low-intensity prescribed fires may not accurately represent "Turbulence from wildfires" a description that could indeed be misleading.

As you correctly pointed out, our data is derived from low-intensity controlled fires rather than wildfires. This choice was made primarily because high-density monitoring of atmospheric turbulence in wildfire conditions is currently very challenging. Existing strategies that measure wildfire-induced turbulence through a few micro-meteorological towers located in the center of the site fail to capture its spatial heterogeneity.

We sought to study turbulence characteristics under different burning conditions in these controlled fires, which, to some extent, resemble those in wildfires. We had further clarified this point to avoid misleading readers into thinking that the study directly addressed large-scale wildfire behavior.

Additionally, we had revised the wording in the introduction to reflect the actual experimental conditions, emphasizing that the study was based on a low-intensity controlled fire environment rather than large-scale wildfires. The title had also been revised to better reflect this context as **Variations in Turbulence Characteristics of Low-Intensity Head Fires Induced by Background Wind and Fuel Loading**.

While this study is based on controlled fire experiments, we believe the results still provide valuable insights into understanding broader fire dynamics. The distinction had been further elaborated in the revised manuscript, clearly delineating the experimental background from the potential scope of its applications.

*4. Some of the language and citation patterns in the introduction are very similar to previously published work, which may give the impression that it lacks originality. While this could be unintentional, a more focused and original discussion of the problem would help the manuscript better address the broader scientific issue and include more relevant research.*

**Response:** We greatly appreciated your feedback regarding the originality and citation patterns in the introduction. In response, the following measures had been taken to improve the manuscript:

a. The introduction had been rewritten to ensure a more concentrated discussion on the core scientific questions addressed by this research. The relationship between fire turbulence characteristics and the fire triangle had been discussed in depth, highlighting the study's focus and clarifying how this research filled gaps in the existing literature.

b. More recent findings relevant to this study had been incorporated into the literature review.

c. Additionally, the language in the introduction had been revised to avoid similarities with previously published works.

*5. Unexplained Data Selection: The manuscript lacks clarity on how the authors selected the subset of data used, which is concerning given that the results are similar to already-published analyses. Providing transparent criteria for data selection would improve the paper's credibility.*

**Response:** Thank you for pointing out this issue. We recognize that providing a clear explanation of the data selection process is critical to the credibility of the research. In response to your suggestion, we have detailed the selection criteria for the data subsets in the manuscript.

As the primary variables of interest in this study are background wind speed and fuel load, efforts were made to control or keep other variables as consistent as possible when selecting the data subsets for the control experiments. These variables included not only fuel conditions (e.g., fuel type, fuel dispersion) and seasonal conditions (e.g., average temperature, humidity, and fuel moisture content), but also fire spread modes (i.e., backing or head fires).

To prevent the fire spread mode from influencing the results, the selected data subsets were exclusively derived from heading fires.

To observe the impact of background wind speed on turbulence characteristics, two data subsets, Burn02 and Burn05, were selected from wind-driven fire experiments with similar seasonal conditions and fuel properties but with significantly different average background wind speeds. The average wind speed of Burn02 was 4.7 m/s, while Burn05 had an average wind speed of 11.1 m/s, more than twice that of Burn02.

To observe the impact of fuel load on turbulence characteristics, the data subsets

Burn11 and Burn12 were selected from wind-driven fire experiments with similar seasonal conditions, the same fuel type, and negligible differences in average wind speed. Burn11 had a fuel load of 0.96 kg/m² , while Burn12 had a fuel load of 1.44 kg/m² , with Burn12 having a fuel load 1.5 times that of Burn11.

*6. Methodology Concerns: The authors claim to have rotated the data into the streamwise coordinate, but this is not evident in the results. After reviewing the provided GitHub code, it seems the rotation step was not properly implemented.*

**Response:** Thank you for your thorough review of the code and methodology. Regarding the issue you raised about the data rotation step not being reflected in the results, we conducted a detailed verification and confirmed that the rotation step was correctly executed. The results after rotation can be verified by checking columns J and K of the data linked at **https://github.com/lcy981009/All-data-generated-during-the-conduct-of-the-study.**

This may be due to the manuscript not providing a sufficiently detailed description of the process, failing to clearly demonstrate the effect of the data rotation; additionally, the redundancy in the code we provided may have led to misunderstanding. Therefore, we took the following corrective measures:

a. Clarification of the rotation process. We have detailed the specific steps of data rotation implementation in the methods section:
Step1:Convert the wind vector represented by the east-west component U(positive for east) and the north-south component V (positive for north) to the format of wind speed and direction:

$$\text{Wind speed} = \sqrt{U^2 + V^2}$$

$$\text{Wind direction} = 180° + (\frac{\arctan 2(-U, -V)}{\pi} \times 180°)$$

The calculated wind direction represents the direction from which the wind comes.
Step2:The prevailing wind direction is calculated based on the average wind direction of all 16 sonic anemometers during the pre-burn period.

Step3: Convert the prevailing wind direction into the form of a unit vector.
The unit vector component in the east-west direction (x-direction):

$$\hat{u}_x = \sin(\theta_{rad})$$

The unit vector component in the north-south direction (y-direction):

$$\hat{u}_y = \cos(\theta_{rad})$$

Where $\theta_{rad}$ is the wind direction angle in radians, the angle needs to be converted from

degrees to radians:

$$\theta_{rad} = \frac{\theta_{\deg} \times \pi}{180}$$

Step4:Calculate the projection of the wind vector in the prevailing wind direction and perpendicular to the prevailing wind direction.
Projection in the prevailing wind direction:

$$\text{Streamwise component} = U \cdot \hat{u}_x + V \cdot \hat{u}_y$$

Projection perpendicular to the prevailing wind direction:

$$\text{Cross-stream component} = U \cdot (-\hat{u}_y) + V \cdot \hat{u}_x$$

b. Results validation. In the results section, we included a comparison of the data before and after rotation, thereby clarifying the impact of this step on the results.

c. Code improvement: We have updated the code on GitHub to make the rotation step clearer and more concise, ensuring its readability.

*7. Lack of In-Depth Analysis: The manuscript summarises the figures without offering sufficient interpretation or explanation of the results. Additionally, the paper does not attempt to relate the findings to previous research in the field, limiting its overall scientific contribution.*

**Response:** We appreciate your valuable comments on the results and discussion section. We recognize the need for a more detailed analysis and interpretation of the results. In response to your suggestions, we have strengthened the discussion by providing more comprehensive figure explanations, highlighting the significance of the results.

Additionally, we also recognize the importance of linking our findings to existing research. In the results and discussion section, the findings of this study have been related to previous research to enhance the scientific contribution of the paper.